# Comparative Chloroplast Genomes and Phylogenetic Relationships of True Mangrove Species *Brownlowia tersa* and *Brownlowia argentata* (Malvaceae)

**DOI:** 10.3390/cimb47020074

**Published:** 2025-01-23

**Authors:** Panthita Ruang-areerate, Duangjai Sangsrakru, Thippawan Yoocha, Wasitthee Kongkachana, Sonicha U-Thoomporn, Onnitcha Prathip Na Thalang, Pranom Chumriang, Poonsri Wanthongchai, Sithichoke Tangphatsornruang, Wirulda Pootakham

**Affiliations:** 1National Center for Genetic Engineering and Biotechnology (BIOTEC), National Science and Technology Development Agency (NSTDA), Pathum Thani 12120, Thailand; panthita.rua@biotec.or.th (P.R.-a.); duangjai.san@biotec.or.th (D.S.); thippawan.yoo@biotec.or.th (T.Y.); wasitthee.kon@biotec.or.th (W.K.); sonicha.uth@biotec.or.th (S.U.-T.); sithichoke.tan@biotec.or.th (S.T.); 2Department of Marine and Coastal Resources, 120 The Government Complex, Chaengwatthana Rd., Thung Song Hong, Bangkok 10210, Thailand; freezpop@hotmail.com (O.P.N.T.); meldec05@gmail.com (P.C.); poonsri56@gmail.com (P.W.)

**Keywords:** *Brownlowia*, Malvaceae, chloroplast genome, plastid, RNA editing, adaptive evolution, phylogenetic analysis

## Abstract

*Brownlowia tersa* and *Brownlowia argentata* are two true mangroves in the genus *Brownlowia* in Malvaceae, and they are a near-threatened and a data-deficient species, respectively. However, the genomic resources of *Brownlowia* have not been reported for studying their phylogeny and evolution. Here, we report the chloroplast genomes of *B. tersa* and *B. argentata* based on stLFR data that were 159,478 and 159,510 base pairs in length, respectively. Both chloroplast genomes contain 110 unique genes and one *infA* pseudogene. Sixty-eight RNA-editing sites were detected in 26 genes in *B. argentata*. A comparative analysis with related species showed similar genome sizes, genome structures, and gene contents as well as high sequence divergence in non-coding regions. Abundant SSRs and dispersed repeats were identified. Five hotspots, *psbI*-*trnS*, *trnR*-*atpA*, *petD*-*rpoA*, *rpl16*-*rps3*, and *trnN*-*ndhF*, were detected among four species in Brownlowioideae. One hotspot, *rps14*-*psaB*, was observed in the two *Brownlowia* species. Additionally, phylogenetic analysis supported that the *Brownlowia* species has a close relationship with *Pentace triptera*. Moreover, *rpoC2* was a candidate gene for adaptive evolution in the *Brownlowia* species compared to *P. triptera*. Thus, these chloroplast genomes present valuable genomic resources for further evolutionary and phylogenetic studies of mangroves and plant species in Malvaceae.

## 1. Introduction

Malvaceae, commonly known as the mallows, is the largest family in Malvales, comprising a diverse group of flowering plants with at least 4225 species in nine subfamilies, namely Bombacoideae, Brownlowioideae, Byttnerioideae, Dombeyoideae, Grewioideae, Helicteroideae, Malvoideae, Sterculioideae, and Tilioideae [1,2]. In the subfamily Brownlowioideae, the genus *Brownlowia* contains approximately 30 species which are widely distributed in Southeast Asia [3,4,5]. Notably, two *Brownlowia* species, *Brownlowia tersa* (Dungun air or the Durian of the sea) and *Brownlowia argentata* (Dungun or Durian Laut), are classified as non-viviparous mangrove species, which grow in inland mangrove forests [3,4,5]. They are known to occur in Borneo and Peninsular Malaysia. *B. tersa* is distributed in Asia including Bangladesh, Brunei, Cambodia, India, Indonesia, Malaysia, Myanmar, Philippines, Singapore, and Thailand [5,6]. *B. argentata* is distributed in Brunei, Indonesia, Malaysia, Myanmar, Papua New Guinea, Philippines, Solomon Islands, and Thailand, and is extinct in Singapore [7]. The two *Brownlowia* species are clearly different in morphological characteristics. For example, *B. tersa* is a shrub that grows to 2/5 meters (m); leaves are narrow and lanceolate are up to 5 centimeters (cm) wide. In contrast, *B. argentata* is a tree up to 10 m; leaves are broad and ovate to 10 cm wide [3,4]. *B. tersa* is commonly used in traditional medicines for the treatment of several symptoms such as boils, diarrhea, dysentery, and wounds because it contains several bioactive compounds that serve for anti-inflammatory, antibacterial, antinociceptive, and antioxidant activities [5,8,9]. It is also used as fuel and materials for house construction in local areas. Based on the IUCN Red List, *B. tersa* and *B. argentata* are listed as a near-threatened (NT) and a data-deficient (DD) species, respectively [6,7,10]. *B. tersa* populations have been decreasing due to habitat loss from shrimp and fish farm construction and coastal development [6,11]; therefore, this species might be at a higher risk of extinction in the future.

Chloroplasts play essential roles in photosynthesis, carbon fixation, and various metabolic pathways in plants. They contain their own genomes with a typically quadripartite structure including a large single copy (LSC), a pair of inverted repeats (IR), and a small single copy (SSC). The size of chloroplast genomes has varied among plant species. In Malvaceae, chloroplast genome sizes range from 143 to 169 kilobases (kb) [12,13,14,15,16,17,18,19,20]. For the subfamily Brownlowioideae, only two species, *Pentace triptera* [16] and *Diplodiscus trichospermus* [18], have had their chloroplast genomes reported. The presence of variations in chloroplast genomes could serve as molecular markers for species identification, phylogenetic analysis, and adaptative evolution [21,22].

Advances in next-generation sequencing technologies have facilitated the exploration of chloroplast genomes in plant species [14,16,18,23,24,25,26]. Recently, single-tube long-fragment read (stLFR) is an efficient technology that enables sequencing from long DNA molecules using second-generation sequencing technology such as BGISEQ-500 and MGISEQ-2000 [27]. The stLFT data is a cost-effective replacement for long reads and could increase accuracy for *de novo* assembly to be near complete and smaller misassembled regions [27]. This technology has been used to complete a chloroplast genome of *Calamus tetradactylus* [26].

Recently, no study has focused on the genome sequence of *Brownlowia* species, which is important for studying evolution and species identification. To generate genetic resources for *Brownlowia* species and to reveal evolutionary relationships of *Brownlowia* and related species, we report the complete chloroplast genome and genetic variation of *B. tersa* and *B. argentata* based on stLFR data. Comparative chloroplast genomes between the two *Brownlowia* species and related non-mangrove and mangrove species were conducted. A phylogenetic tree among forty-one plant species within Malvaceae and three outgroups was performed. Moreover, genes under purifying selection or positive selection in the two *Brownlowia* species were identified to investigate whether genes related to adapt in environments of inland coastal areas. Therefore, these *Brownlowia* chloroplast genomes provide valuable insights into their genomic features, evolutionary phylogenetic relationships, and gene selective pressures in the subfamily Brownlowioideae in Malvaceae.

## 2. Materials and Methods

### 2.1. Plant Materials, DNA Extraction, and Sequencing

Young fresh leaves from a mature *Brownlowia tersa* (L.) Kosterm (N 8.280251, E 98.737898) and a mature *Brownlowia argentata* Kurz (N 9.233155, E 99.238231) were collected in 2023 from the mangrove forest in Ao Luek Noi District in Krabi Province and Phunphin District in Surat Thani Province, Thailand, respectively, under the supervision of Thailand’s Department of Marine and Coastal Resources. They were flash-frozen and maintained in liquid nitrogen until use. Genomic DNA in each sample was extracted from the young leaves using CTAB [28]. Purified DNA was used to generate stLFR sequencing library that was constructed following the protocol of the MGIEasy stLFR Library Prep Kit (MGI, Shenzhen, China), and sequenced on MGISEQ-2000RS (MGI, Shenzhen, China) in house.

### 2.2. Chloroplast Genome Assembly and Annotation

The stLFR reads of *B. tersa* and *B. argentata* were used for assembling the *Brownlowia* chloroplast genomes using GetOrganelle v.1.4.1 [29] based on a reference genome-based strategy with *Diplodiscus trichospermus* from the GenBank sequence database under accession number OP572286 following a pipeline in Appendix A. These *Brownlowia* chloroplast genomes were then annotated using GeSeq online tool v.2.03 [30]. The start–stop loci and intron–exon junctions of coding genes were manually examined by comparing them with the chloroplast genomes of *D. trichospermus* and other mangrove species. Apart from the coding genes, transfer RNAs (tRNAs) were identified using ARAGORN v1.2.38 [31] and tRNAscan-SE v2.0.3 [32] in the GeSeq software v.2.03. Circular chloroplast genomes were visualized using OGDRAW v1.3.1 [33]. Finally, the annotated chloroplast sequences of *B. tersa* and *B. argentata* were deposited in GenBank with accession numbers PP419968 and PP419969, respectively.

### 2.3. RNA Sequencing and RNA Editing Site Identification

For RNA sequencing, RNA from leaf tissues of the two *Brownlowia* species in the same sample collection was extracted following the protocol in Pootakham et al. (2021) [34]. Poly(A) mRNA was subsequently purified using the Dynabeads mRNA purification Kit (ThermoFisher Scientific, Waltham, Massachusetts, USA). The integrity of RNA was evaluated using the Fragment Analyzer System (Agilent, Santa Clara, CA, USA). Libraries were constructed in each sample with 200 ng of poly(A) mRNA using the MGIEasy RNA Library Prep Kit V3.0 (MGI Tech, Shenzhen, China) and were sequenced using the MGISEQ-2000RS Sequencing Flow Cell v3.0 (MGI Tech, Shenzhen, China). The short-read RNA sequences of *B. tersa* and *B. argentata* were obtained and deposited in sequence read archive (SRA) with SRR31035998 and SRR31066870, respectively.

To identify candidate RNA editing sites in all protein-coding genes of both *Brownlowia* species, the RNA sequences of each species were mapped to each chloroplast genome using Burrows–Wheeler aligner (BWA) v.0.7.17-r1188 with default parameters [35]. A sequence alignment map (SAM) file was converted to a binary alignment map (BAM) file using samtools v1.9 [36]. The index of BAM files was also created using samtools v1.9. To examine RNA-editing sites of the genes, the alignment results of BAM files were visualized using integrative genomics viewer (IGV) v2.17 [37]. RNA-editing sites where cytidine (C) to uridine (U) conversion events were present were identified with read coverage by at least 30 reads and editing events by at least 20%.

### 2.4. Chloroplast Genome Comparative Analysis

Chloroplast genome comparative analysis in eight plant species in Malvaceae including two *Brownlowia* species, two reported plant species in Brownlowioideae (*Pentace triptera*: BGT7001 and *D. trichospermus*: OP572286), two mangrove species in Sterculioideae (*Heritiera fomes*: OM022247 and *Heritiera littoralis*: NC_043923), and two mangrove associates in Malvoideae (*Talipariti tiliaceum*: MN533969 and *Thespesia populnea*: NC_048518) was performed using mVistra with the Shuffle-LAGAN mode [38]. The chloroplast sequence of *P. triptera* is available in DRYAD (https://datadryad.org/stash/share/LkbLwUlzW_GJ5rBDMHYZz69S19HzkNWY6fPOySH9tBQ, accessed on 2 May 2024) [16], whereas *D. trichospermus* is available in the GenBank database. The chloroplast genome of *B. tersa* was used as a reference for comparison. In addition, the junctions and borders of the IR regions among the eight plant species were visualized using the CPJSdraw software v1.0.0 [39].

### 2.5. Nucleotide Diversity Analysis

Two datasets were used to estimate nucleotide diversity (Pi). The first dataset was four species, *B. tersa*, *B. argentata*, *P. triptera*, and *D. trichospermus*. The second dataset was only two *Brownlowia* species. Both protein-coding genes and interspace regions of the two datasets were aligned in each region using MUSCLE. The nucleotide diversity values (Pi) in each aligned sequence region were calculated using DnaSP version 6.12.03 [40].

### 2.6. Analysis of Simple Sequence Repeats and Dispersed Repeats

Simple sequence repeats (SSRs) were identified among the four chloroplast genomes in Brownlowioideae, including *B. tersa*, *B. argentata*, *P. triptera*, and *D. trichospermus*, using MISA v2.1 [41]. The identified SSRs included mononucleotide repeats ≥ 10 bases, dinucleotides ≥ 10 bases (five repeats), trinucleotides ≥ 12 bases (four repeats), tetranucleotides ≥ 12 bases (three repeats), pentanucleotide ≥ 15 bases (three repeats), and hexanucleotides ≥ 18 bases (three repeats). Moreover, dispersed repeat sequences, including complement, forward, palindrome, and reverse, were identified using REPuter online tool (https://bibiserv.cebitec.uni-bielefeld.de/reputer/, accessed on 6 August 2024) [42].

### 2.7. Phylogenetic Analysis

To confirm the position of the *Brownlowia* species, a phylogenetic analysis of 44 plant species was performed using a maximum likelihood (ML) method based on 68 shared protein-coding genes (Appendix A). The 44 plant species were four species in Brownlowioideae (*B. argentata*, *B. tersa*, *P. triptera*, and *D. trichospermus*), thirty-seven plant species in other subfamilies in Mavalceae, and three outgroup species (*Hopea hainanensis*: MN533970, *Vatica mangachapoi*: NC_041485, and *Vativa odorata*: MN533976). The 68 conserved genes include *accD*, *atpA*, *atpB*, *atpE*, *atpF*, *atpH*, *atpI*, *ccsA*, *cemA, clpP, matK*, *ndhA*, *ndhB*, *ndhC*, *ndhD*, *ndhE*, *ndhG*, *ndhH*, *ndhI*, *ndhJ*, *petA*, *petB*, *petD*, *petG*, *petL*, *petN*, *psaB*, *psaC*, *psaI*, *psaJ*, *psbA*, *psbC*, *psbD*, *psbE*, *psbF*, *psbH*, *psbI*, *psbJ*, *psbK*, *psbL*, *psbM*, *psbN*, *psbT*, *rbcL*, *rpl2*, *rpl14*, *rpl16*, *rpl20*, *rpl23*, *rpl32*, *rpl33*, *rpl36*, *rpoA*, *rpoB*, *rpoC1*, *rpoC2*, *rps2*, *rps3*, *rps4*, *rps7*, *rps8*, *rps11*, *rps12*, *rps14*, *rps15*, *rps18*, *ycf3*, and *ycf4*. Each gene sequence was aligned using MUSCLE with default in MEGA X [43], and the aligned sequences were concatenated in each species. The GTR + I + G model as the best-fit model was identified using the best DNA/protein model tool in MEGA X. A maximum likelihood (ML) analysis was used to construct a phylogenetic tree using RAxML version 8.2.10 [44] with a GTRGAMMAI (GTR + I + G) model. Node support was estimated by performing 1000 bootstrap replicates. Finally, the phylogenetic tree was visualized using FigTree v1.4.3 (http://tree.bio.ed.ac.uk/software/figtree/, accessed on 23 August 2024). In addition, all chloroplast genomes were blasted against with an *infA* gene of *T. tiliaceum* (LC605014) to check this gene in their chloroplast genomes. Then, the chloroplast *infA* gene loss and gain in plant species in Malvaceae were plotted onto clades of the phylogenetic tree.

### 2.8. Gene Selective Pressure Analysis

The sixty-eight shared chloroplast protein-coding genes were used to investigate selection pressures for the two *Brownlowia* species. Compared species pairs contained between each *Brownlowia* species and two terrestrial plant species in Brownlowioideae (*P. triptera* and *D. trichopermus*). Each protein-coding gene that was translated to amino acid sequences was aligned using MEGA X [43]. The ratios of the rate of non-synonymous substitutions (Ka) to the rate of synonymous substitutions (Ks) or Ka/Ks ratios of each gene in each species pair were calculated using KaKs-calculator v2.0 [45]. R with the heatmap function [46] was used to visualize the Ka/Ks ratios of genes in each species pair. Genes with Ka/Ks ratios greater than 1 were determined to be under position selection, whereas genes with Ka/Ks ratios less than 1 were determined to be under purifying selection. Notably, genes with Ka/Ks ratios ≥ 40 or not available (NA) were replaced and set to zero because they had extremely low Ks values or no substitution, respectively.

Furthermore, genes with Ka/Ks ratios > 1 were tested statistically with the Chi-square test to identify potential positive selection sites across lineages using the codeml program from PAML v4.9 [47] based on the branch–site model [48]. The *Brownlowia* species and others were set as a foreground and background branches, respectively.

## 3. Results

### 3.1. Characteristics of B. tersa and B. argentata Chloroplast Genomes

Two chloroplast genomes of *Brownlowia* were assembled based on stLFR reads. A total of 13.3 and 14.2 Gb stLFR raw reads were used for the chloroplast genome assemblies of *B. tersa* and *B. argentata*, respectively. The chloroplast genome sizes of *B. tersa* (159,478 bp) and *B. argentata* (159,510 bp) were similar, and GC contents were all 37.05% (Figure 1 and Table 1). They exhibited a conserved quadripartite structure, consisting of a large single copy (LSC: 88,394–88,435 bp, 34.89% GC content), a small single copy (SSC: 19,984–19,985 bp, 31.35–31.36% GC content), and two inverted repeats (IRs: 25,545–25,550, 43.01% GC content) (Table 1). Each chloroplast genome contained 130 genes, with 84 protein-coding genes, 8 ribosomal RNAs (rRNAs), 37 transfer RNAs (tRNAs), and 1 *infA* pseudogene. Among those genes, three genes (*clpP*, *rps12* and *ycf3*) had two introns, and nine genes (*atpF*, *ndhA*, *ndhB*, *petB*, *petD*, *rpoC1*, *rpl2*, *rpl16*, and *rps16*) had one intron (Table 2). Notably, the *rps12* gene was trans-spliced because of the location of the first exon at the LSC and the other two exons at the IRs (Figure 1). In total, 16 genes were duplicated in the IRs, including 5 protein-coding genes (*ndhB*, *rpl2*, *rpl23*, *rps7*, and *ycf2*), 4 rRNA genes (*rrn4.5*, *rrn5*, *rrn16*, and *rrn23*), and 7 tRNA genes (*trnA-UGC*, *trnI-CAU*, *trnI-GAU*, *trnL-CAA*, *trnN-GUU*, *trnR-ACG*, and *trnV-GAC*). One *infA* pseudogene was identified in the two *Brownlowia* species due to partial deletion (1–22 bp).

### 3.2. Comparative Chloroplast Genomes

Comparing chloroplast genomes between two *Brownlowia* species and six plant species in Malvacaea including two non-mangrove chloroplast genomes in Brownlowioideae (*P. triptera* and *D. trichospermus*) and four mangrove chloroplast genomes in Malvaceae (*H. littoralis*, *H. fomes*, *T. populnea*, and *T. tiliaceum*), chloroplast genomes, GC contents, and gene contents were relatively similar (Table 1 and Appendix A). The size of the chloroplast genomes ranged from 158,570 (*D. trichospermus*) to 168,778 (*H. littoralis*) bp in length, and all had ~37% GC contents. They contained 128−132 genes, including 83−86 protein-coding genes, 8 rRNAs, 36−37 tRNAs, and 1 *infA* pseudogene.

The differences among the eight chloroplast genomes were also evaluated using mVISTA, with the annotated *B. tersa* chloroplast genome as a reference (Figure 2). The chloroplast genomes of the two *Brownlowia* species were highly conserved with non-mangrove species in Brownlowioideae (*P. triptera* and *D. trichospermus*) when compared with other mangroves species in Malvaceae. IR sequences in all chloroplast genomes had a low divergence level. High levels of divergence between the *Brownlowia* species and other species were mainly concentrated in non-coding regions of LSC and SSC regions such as *trnH-psbA*, *petN-psbM*, *ycf3-trnS*, *trnT-trnF*, *trnT-psbD*, *ndhF-rpl32*, and *rpl32-trnL*.

In addition, the comparison of the junctions of the LSC, SSC, IRa, and IRb among eight chloroplast genomes in Malvaceae are presented in Figure 3. The SSC orientation of *B. tersa* was designed as a reference. By aligning the genomes to the reference sequence, the SSC sequence in most species has a forward-read orientation, whereas *P. triptera* possesses a reverse orientation. *Rpl2* and *trnH* genes were found at the IRa/LSC boundary in all chloroplast genomes. The *rpl2* gene is located in the IRa region and the *trnH* gene is located in the LSC region, 0–2 bp from the IRa/LSC boundary. The other boundaries of the *Heritiera* species were different from those of the others due to their long IR regions. At the junction of LSC/IRb, *rps19* was found in the *Brownlowia* species and other four species, *P. triptera*, *D. trichospermus*, *T. tiliaceum*, and *T. populnea*. It was located in the IRb region with 3 bp (*T. tiliaceum* and *T. populnea*), 6 bp (*B. tersa*, *B. argentata* and *P. triptera*) and 8 bp (*D. trichospermus*) interval after the LSC/IRb boundary. The *ycf1* gene crossed the SSC and IR boundary with 33 bp (*B. tersa* and *B. argentata*), 36 bp (*P. triptera*), 78 bp (*T. populnea*), 109 bp (*D. trichospermus*) and 960 bp (*T. tiliaceum*) of the 3′ end of this gene in the IR region. The *ndhF* gene was located in the SSC region with a 39 bp (*B. tersa* and *B. argentata*), 51 bp (*D. trichospermus*), 56 bp (*P. triptera*), 175 bp (*T. populnea*), and 330 bp (*T. tiliaceum*) interval to the boundary of the IR and SSC. The *rpl2* and *trnN* genes were located in IR regions near the LSC/IRs junction and the SSC/IRs junction in all species in Brownlowioideae and *T. populnea*, respectively.

### 3.3. RNA Editing Sites in Chloroplast Genes of Brownlowia

In the *B. argentata* chloroplast genome, there are 68 C-to-U sites in 26 protein-coding genes with 9, 56, and 3 sites at the first, second, and third codon positions, respectively (Table 3). The RNA-editing sites resulted in sixty-five nonsynonymous and three synonymous editing sites. A majority of editing events were observed in codon changes in the second position, including CCU to CUU and CCA to CUA (Pro to Leu), UCU to UUU (Ser to Phe), UCA to UUA and UCG to UUG (Ser to Leu), and ACG to AUG (Thr to Met). Approximately half of the amino acid changes from the RNA-editing events were Ser to Leu. In addition, the start codon of the chloroplast *ndhD* gene had the RNA editing event, an ACG codon (84%) to an ATG codon (16%). Of these editing events, 86% of the first-position edits consisted of His (CAC and CAU) to Tyr (UAC and UAU) codon changes. All the third-position editing sites were nonsynonymous UUC to UUU (Phe to Phe) and GUC to GUU (Val to Val). Exhibiting an editing efficiency of up to 80% was found in thirteen genes including *atpA*, *atpE*, *atpI*, *psbZ*, *rps14*, *accD*, *psbJ*, *psbF*, *petL*, *clpP*, *pdbN*, *petB*, and *ndhD*. Nevertheless, no RNA editing sites were detected in *B. tersa* based on the leaf transcriptome set in this study.

### 3.4. Simple Sequence Repeats and Dispersed Repeats

To identify genetic variation between *Brownlowia* species and two related species in Brownlowioideae, simple sequence repeats (SSRs) and dispersed repeats were examined (Figure 4, Appendix A). For SSRs, a total of 85, 83, 82, and 81 SSRs were identified in *B. tersa*, *B. argentata*, *P. triptera*, and *D. trichospermus*, respectively (Figure 4A and Appendix A). The most abundant of SSR types were mononucleotide repeats, accounting for 74.22–79.17%. Remaining SSRs included 9–10 tetranucleotide repeats (8.91–10.30%), 3–8 dinucleotide repeats (3.09–7.92%), 3–8 trinucleotide repeats (3.12–8.24%), 2–4 pentanucleotide repeats (1.98–4.12%), and 1 hexanucleotide repeat in *P. triptera*. These SSRs were mostly located in the LSC region rather than other regions (Figure 4B). For example, 66, 8, and 11 SSRs were discovered within the LSC, SSC, and IR regions of *B. tersa*, respectively. The A/T tandem repeat was most frequent (70.10–76.04%) (Figure 4C). Notably, a few unique A/T tandem repeats were detected between *B. tersa* and *B. argentata* (Appendix A). The second most abundant SSRs were AAAT/ATTT, ranging from five in *B. tersa*, *B. argentata*, and *P. triptera* to six in *D. trichospermus*. It is noteworthy that AACT/AGTT, AGAT/ATCT, and AAAAT/ATTTT repeats were detected in three species, *B. tersa*, *B. argentata*, and *P. triptera*, whereas AATG/ATTC and AAAAG/CTTTT repeats were detected in only *D. trichospermus*. The AAAGAT/ATCTTT repeat was detected only in *P. triptera*.

Furthermore, dispersed repeats (F: forward; R: reverse; P: palindromic; and C: complement) were detected in four chloroplast genomes in Brownlowioideae (Figure 4D and Appendix A). Each chloroplast genome contained 49 dispersed repeats. Although the numbers of dispersed repeats were similar, they were different in the four species. There are 17, 17, 13, and 16 forward repeats, 10, 11, 14, and 13 reverse repeats, 22, 21, 19, and 20 palindromic repeats, and 0, 0, 3, and 0 complement repeats for *B. tersa*, *B. argentata*, *P. triptera*, and *D. trichospermus*, respectively. Among the species, palindromic repeat (38.78–44.90%) was the most common type, followed by forward repeat (26.53–34.69%), and reverse repeat (20.41–28.57%). Complement repeat was detected in only *P. triptera*. Almost all repeats were in the range of 21–30 bp in length (Figure 4E). The number of dispersed repeats with a unit length of <20 bp in *D. trichospermus* (13, 26.53%) was higher than those in other species (1–2, 2%). Most dispersed repeats were located in the LSC and IR regions (Figure 4F). In the SSC region, dispersed repeats were found in only *D. trichospermus*.

### 3.5. Nucleotide Diversity

In the two datasets, including four species in Brownlowioideae (*B. tersa*, *B. argentata*, *P. triptera*, and *D. trichospermus*) and two *Brownlowia* species, the nucleotide diversity (Pi) of 68 protein-coding genes and 123 interspace regions was estimated (Figure 5). The Pi value of the protein-coding genes ranged from 0 to 0.011 in the four species and 0 to 0.010 in the two *Brownlowia* species (Figure 5A). Most Pi values in the two *Brownlowia* species were 0. The Pi value of *psbM* in the LSC region of *Brownlowia* (Pi = 0.010) showed the highest value, which was higher than the Pi value of *psbM* in the four species (Pi = 0.005). In addition, the Pi value of the interspace regions ranged from 0 to 0.048 in the four species and 0 to 0.039 in the two *Brownlowia* species (Figure 5B). Most Pi values in the two *Brownlowia* species also were 0. Interestingly, the Pi value of *rps14*-*psaB* (Pi = 0.039) in two *Brownlowia* species was higher than the Pi value of *rps14*-*psaB* in the four species (0.020). In the four species, there were five interspace regions, *psbI*-*trnS*, *trnR*-*atpA*, *petD*-*rpoA*, *rpl16*-*rps3*, and *trnN*-*ndhF*, that passed the threshold Pi > 0.25. Four interspace regions, *psbI-trnS*, *trnR*-*atpA*, *petD*-*rpoA*, and *rpl16*-*rps3*, were located in the LSC region, while *trnN*-*ndhF* was located in the IR region. The highest Pi value (Pi = 0.048) was found in the *petD*-*rpoA* region, and the second-highest Pi value (Pi = 0.044) was found in the *rpl16*-*rps3* region.

### 3.6. Phylogenetic Relationships

A maximum likelihood (ML) was constructed based on 68 conserved protein-coding genes among 41 plant species in nine subfamilies in Malvaceae and three outgroup species. (Figure 6). There were nine clades in Malvaceae including representative nine subfamilies. The ML tree showed the existence of two major clades, Malvadendrina and Byttneriina. The Malvadendrina clade contained seven subfamilies including Bombacoideae, Brownlowioideae, Dombeyoideae, Helicteroideae, Malvoideae, Sterculioideae, and Tilioideae. The Byttneriina contained two subfamilies including Byttnerioideae and Grewioideae. In Brownlowioideae, the phylogenetic tree shows that two *Brownlowia* species and two terrestrial plant species (*P. triptera* and *D. trichospermus*) are monophyletic. *B. tersa* and *B. argentata* were clustered together and *P. triptera* was a sister group, with 100% bootstrap support. These *Brownlowia* species are closely related to the terrestrial species *P. triptera* than *D. trichospermus*. In addition, Brownlowioideae formed a sister clade with the clade of Dombeyoideae and Tilioideae, with 88% bootstrap support. Moreover, chloroplast *infA* gene loss and gain in Malvaceae were examined. The *infA* pseudogene was found in the *Brownlowia* species and other most plant species in Malvaceae, while the *infA* gain was observed in the genus *Althaea, Hibiscus*, *Talipariti*, and *Tilia*.

### 3.7. Selective Pressure Genes

The Ka/Ks ratios of 68 shared protein-coding genes were calculated between pairwise species of four species in Brownlowioideae (BA—*B. argentata*; BT—*B. tersa*; PT—*P. triptera*; and DT—*D. trichospermus*) (Figure 7 and Appendix A). *D. trichospermus* was assumed to be an ancestor of Brownlowioideae based on the result of our phylogenetic tree. Among species pairs, the Ka/Ks ratios in the chloroplast genes were around 0.07, indicating strong purifying selections. In particular, *rpoC2* had Ka/Ks ratios > 1.0 in two pairwise species, BA-PT (the Ka/Ks ratio = 2.32) and BT-PT (1.11), implying possible positive selection in the *Brownlowia* species compared to *P. triptera*. To evaluate positive sites based on Bayes Empirical Bayes (BEB) analysis, *rpoC2* (ω > 1) was positively selected in five amino acid sites/changes (80L: Leu (PT) to Phe (BA and BT); 594M: Met to Ile; 697Q: Gln to Pro; 894H: His to Arg; and 1324R: Arg to Trp).

## 4. Discussion

### 4.1. Chloroplast Genome Structure and Evolution in the Genus Brownlowia

In this study, the chloroplast genomes of *B. tersa* and *B. argentata* were successfully assembled into 159.5 kb based on stLFR data. The chloroplast genome size of the two *Brownlowia* species is consistent with related land plant species in Brownlowioideae such as *Pentace triptera* (159.4 kb) and *Diplodiscus trichospermus* (158.6 kb) [16,18]. It was slightly smaller than the chloroplast genome size of other true mangroves (*Heritiera angustata*, *Heritiera fomes*, and *Heritiera littoralis*) and mangrove associates (*Thespesia populnea* and *Hibiscus*/*Talipariti tiliaceum*) in Malvaceae as well as mangroves in Combretaceae, Euphorbiaceae, and Rhizophoraceae, whose size varied from 160.1 kb to 168.8 kb in length [12,15,17,19,24,25]. On the contrary, the size of *Brownlowia* genomes was larger than the size of mangrove chloroplast genomes in Acanthaceae (148.3–150.3 kb) [15,24,49]. The chloroplast genome sizes generally varied among different plant species due to the contraction, expansion, and loss of IR regions.

Comparative analyses of the two *Brownlowia* species and representatives of two other genera of Brownlowioideae (*Pentace* and *Diplodiscus*) revealed a similar pattern of structure and gene order as well as an equal number of genes. However, high levels of divergence between *Brownlowia* and *Diplodiscus* were observed in several intergenic spacer regions, which is concordance with the phylogenetic result in this study showing that *Diplodiscus* was less closely related species than *Pentace*. IR regions had lower divergence and were more conserved compared to LSC and SSC regions, which is consistent with the result of the nucleotide diversity analysis.

Among plant species, the contraction and expansion of IRs is a common event leading to gene gain and loss in IR regions [50,51]. In our study, complete *rps19* and *ycf1* genes among four species in Brownlowioideae were found adjacent to the junctions, with slight length variations. For instance, the full *ycf1* gene crossed the SSC/IRa with short length variations (33 bp in *Brownlowia* to 109 bp in *Diplodiscus*) of the 3′ end of this gene in the IRa region, indicating a slight contraction of the IR region in the *Brownlowia* species compared to other species in Malvaceae. The contraction and expansion of IR regions are main factors for the genome length and number of genes in the chloroplast genomes of each plant species during evolution.

### 4.2. RNA Editing in Brownlowia Species

Chloroplast RNA editing events in higher plants mainly occur through the conversion of cytidine to uridine (C to U) [52]. RNA editing is essential for various plant developmental processes and evolutionary adaptation such as plant embryogenesis, growth, adaptation to environmental changes, and signal transduction [53,54]. In the present study, no RNA editing site was detected in *B. tersa* based on one transcriptome dataset. The absence of RNA editing can cause abnormal plant development such as etiolation and yellow leaves [55,56,57]. Therefore, more leaf RNA data should be generated to evaluate whether RNA editing sites exist in the *B. tersa* chloroplast genome. In *B. argentata*, most RNA-editing sites were found at the second codon positions, followed by the first and third positions, which is consistent with several terrestrial plant species [58,59,60] and mangrove species in Rhizophoraceae [61]. Our editing events at the second codon positions are affected by amino acid changes, especially Ser-to-Leu, Pro-to-Leu, and Ser-to-Phe. These events were found in several plant species [58,59,60,61]. Notably, *ndhB* involving cyclic electron flow around photosystem I had the largest number of RNA editing sites in *Brownlowia* species and other plant species [60,61,62]. In particular, the start codon of *ndhD* in *Brownlowia* species was affected by an RNA-editing event from ACG to AUG with 16% editing efficiency, which is consistent with ACG codon rather than AUG codon at a translation initiation site in other plant species [63,64,65]. In fact, the ACG codon could be the initiation codon for translation in *ndhD* in some species [66]. The ACG-to-AUG editing varies and depends on developmental and environmental factors [63]. In addition, some RNA editing sites were unique to plant species, suggesting adaptive evolution. For example, RNA-editing sites in several genes such as *atpA*, *ndhA*, *rpl20*, *rpl23*, *rpoC1*, and *rps1* were observed in *B. argentata*, but not in other mangrove species in Rhizophoraceae [61]. These genes might be candidate genes for studying the adaptive evolution in mangrove *Brownlowia* species.

### 4.3. SSRs, Dispersed Repeats, and Nucleotide Diversity

To identify differences in the chloroplast genomes of two *Brownlowia* species and two land plant species in Brownlowioideae, SSRs and repeats were examined. Mononucleotides have the most SSR repeats that are common in plants [24,25,67]. Different SSRs may be produced as genetic markers for identifying *Brownlowia* species and related species. Dispersed repeats were observed with the highest number of palindromes, followed by forward and reverse types, which are in concordance with several plants [68,69]. On the contrary, forward repeat is a major type of dispersed repeat in many mangrove chloroplast genomes [24,25,49]. Indeed, both palindromic and forward repeats accounted for a high proportion in chloroplast genomes compared to reverse and complementary repeats [24,25,49,68,69]. The high peak of Pi diversity among four species in Brownlowioideae was observed in non-coding regions (five hotspot regions). High Pi values are generally found in spacer regions between genes [70]. These genetic variations are important as resources of molecular markers and evolution studies.

### 4.4. Phylogenetic Relationships of Malvaceae

Phylogenetic reconstructions based on 68 chloroplast protein-coding genes among 41 species in nine subfamilies in Malvaceae and three outgroup species have provided insights into the evolutionary history of *Brownlowia* species. Our ML tree and several previously published trees in Brownlowioideae had highly congruent topologies [2,71,72,73]. For example, the phylogenetic relationships based on a chloroplast *ndhF* sequence, *Brownlowia, Pentace, Berrya*, and *Carpodiptera* were a monophyletic group in the clade Brownlowioideae [71]. The phylogenetic relationships with four chloroplast genes, *atpB*, *matK*, *ndhF*, and *rbcL*, also showed the placement of *Brownlowia* together with *Pentace* and *Berrya* in the clade Brownlowioideae [72]. The Bayesian phylogenetic tree with eight molecular markers (six chloroplast genes: *rbcL*, *atpB, trnK-matK*, *trnH*-*psbA*, t*rnL*-t*rnF*, and *ndhF*; one mitochondrial *matR* gene; and one nuclear internal transcribed spacer (ITS) sequence) agreed that *Brownlowia* was closely related to *Pentace*, followed by *Diplodiscus, Jarandersonia, Berrya, Carpodiptera and Chritiana* in the clade Brownlowioideae [73]. Thus, the relationships of *Brownlowia* species employing chloroplast genes are definitively resolved with the best-supported topology.

Resolving relationships among subfamilies in Malvaceae are critical for evaluating their genetic evolution. There were numerous phylogenetic analyses in Malvaceae [2,16,18,62,71,72,73,74]. The phylogenetic positions of subfamilies with Malvaceae were ambiguous. However, the subfamilial relationships in Malvaceae have been resolved recently using whole chloroplast genes or chloroplast genomes [16,18]. Our phylogenetic tree confirmed two major clades of Malvaceae, Byttneriina, and Malvadendrina. The clade of Byttneriina containing two subfamilies, Byttnerioideae and Grewioideae, were sister to the remainder clades of Malvadendrina, with seven subfamilies, including Brownlowioideae, Bombacoideae, Dombeyoideae, Helicterroideae, Malvoideae, Sterculioideae, and Tiliodeae [2,16,18,62,71,72,73,74]. Helicterroideae occupied the earliest branching position within Malvadendrina based on chloroplast genes [16,18,62,71,73]. In contrast, Dombeyoideae based on an ITS sequence [72] or Tilioideae based on chloroplast *atpB* and *rbcL* sequences [2] were located at the base of Malvadendrina. The present study showed the clade of Brownlowioideae that was clearly separated from the clade of Tilioideae with a high support value. Dombeyoideae and Tilioideae were grouped together and were sister to Brownlowioideae, which is in concordance with recently previous phylogenetic relationships in Malvaceae based on numerous chloroplast genes [16,18]. Malvoideae and Bombacoideae (Malvatheca) formed a close clade with high confidence (100% bootstrap value), which is in concordance with previous studies [2,16,18,71]. In our newly phylogenetic tree, Sterculioideae and Malvatheca were grouped together with a 54% bootstrap value (no high support). It is consistent with some studies [72,73], while it is contrast to the resolved phylogenetic trees that Sterculioideae formed a clade sister to Malvatheca and Brownlowioideae, Tilioideae, and Dombeyoideae [16,18]. In fact, there was still difficulty over taxonomic divisions based on morphological features within these families [75]. For example, androecial traits shared by Sterculioideae and Malvatheca, such as sessile or subsessile anthers and the lack of staminodes, support their sister group relationships [72]. These results from our phylogenetic relationships and others reveal that numerous chloroplast gene sequences could resolve phylogenetic relationships among subfamilies in Malvaceae that led to reliable phylogenetic positions. Many genetic variations and morphological data should be compared and combined to evaluate phylogenetic relationships among subfamilies in Malvaceae.

Among protein-coding genes, the *infA* pseudogene was found in *Brownlowia* and most plants in Malvaceae, except *Althaea, Hibiscus*, *Talipariti*, and *Tilia* species that have a complete *infA* gene [12]. The *infA* gene could transfer from chloroplast to nuclear during angiosperm evolution [76].

### 4.5. Selective Pressure

Challenging environments may impose selective pressure on chloroplast genes that are involved in photosynthesis adaptation to extreme environments. In our study, we found positive selection in *rpoC2* between *Brownlowia* and *Pentace* species, indicating adaptive evolution in *Brownlowia* species. It was also under positive selection in various plant species growing in divergent environments such as light intensity, high altitude, and coastal ecotype [77,78,79,80,81]. The *rpoC2* gene encodes RNA polymerase subunit that contained the large number of non-synonymous sites compared with other *rpo* genes (*rpoA*, *rpoB*, and *rpoC1*) and was associated with the structural properties of the respective plastid-encoded polymerase subunits under positive pressure [77]. Thus, the *rpoC2* gene in *Brownlowia* species may be adapted to inland coastal areas.

## 5. Conclusions

In this study, we newly sequenced and reported the chloroplast genome of two true mangroves in the genus *Brownlowia*, *Brownlowia tersa* (a near-threatened species) and *Brownlowia argentata* (a data deficient species), based on stLFR data. Our study also revealed chloroplast structure, protein-coding genes, RNA editing sites, repeats, and SSRs. Chloroplast genome structure, gene content, and gene organization of the two *Brownlowia* species were similar. Notably, the chloroplast genomes may not be efficient markers for DNA barcoding in *Brownlowia* mangrove species due to high genetic similarity. Among *B. tersa*, *B. argentata*, *Pentace triptera*, and *Diplodiscus trichospermus* in Brownlowioideae, specific SSRs and hotspot regions such as *psbI*-*trnS*, *trnR*-*atpA*, *petD*-*rpoA*, *rpl16*-*rps3*, and *trnN*-*ndhF* could be utilized as potential markers. Furthermore, a phylogenetic tree based on 68 conserved chloroplast protein-coding genes across plant species in nine subfamilies in Malvaceae showed that *B. tersa* and *B. argentata* have closer relationships with *P. triptera*. Among genes, *rpoC2* was subjected to positive selection. These results provide valuable insights into studying evolution and phylogenetic relationships in *B. tersa* and *B. argentata* and their related species.

## Figures and Tables

**Figure 1 cimb-47-00074-f001:**
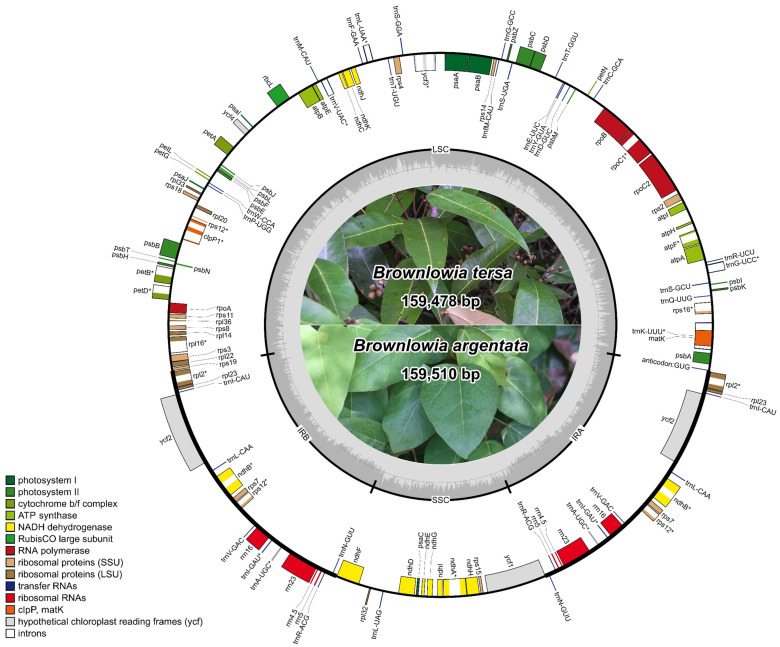
The circular chloroplast maps of *Brownlowia tersa* and *Brownlowia argentata*. Genes located inside and outside the circle are transcribed counter-clockwise and clockwise, respectively. In the inner circle, the gray and lighter gray areas indicate GC and AT contents of the chloroplast genome. LSC is a large single copy. IRA and IRB are two inverted repeats. SSC is a small single copy. Introns in genes are indicated by an asterisk symbol (*). Genes are shown in different colors based on different functional groups.

**Figure 2 cimb-47-00074-f002:**
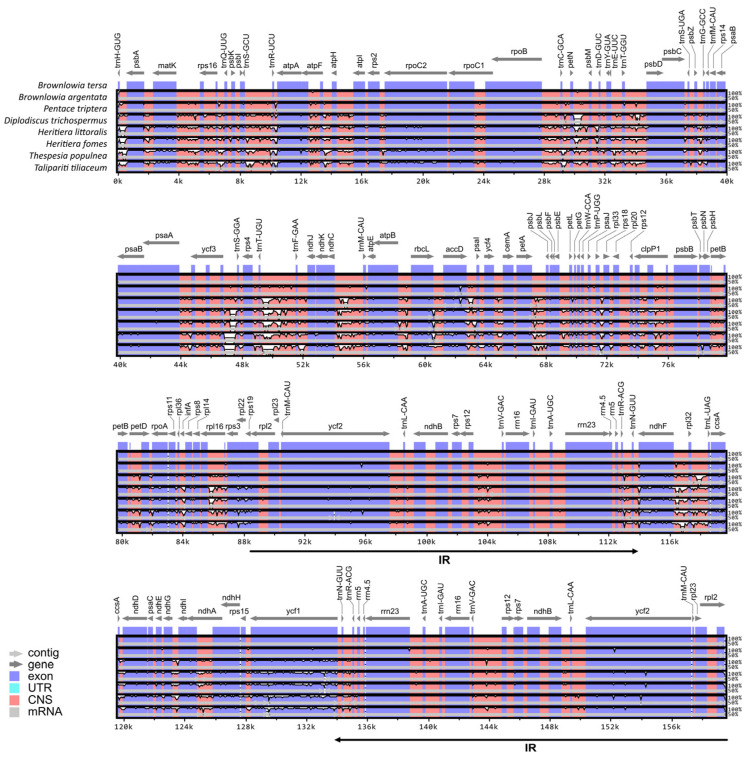
Alignment map of eight chloroplast genomes in Malvaceae. Vertical and horizontal axes represent a percentage sequence identity and genome position, respectively.

**Figure 3 cimb-47-00074-f003:**
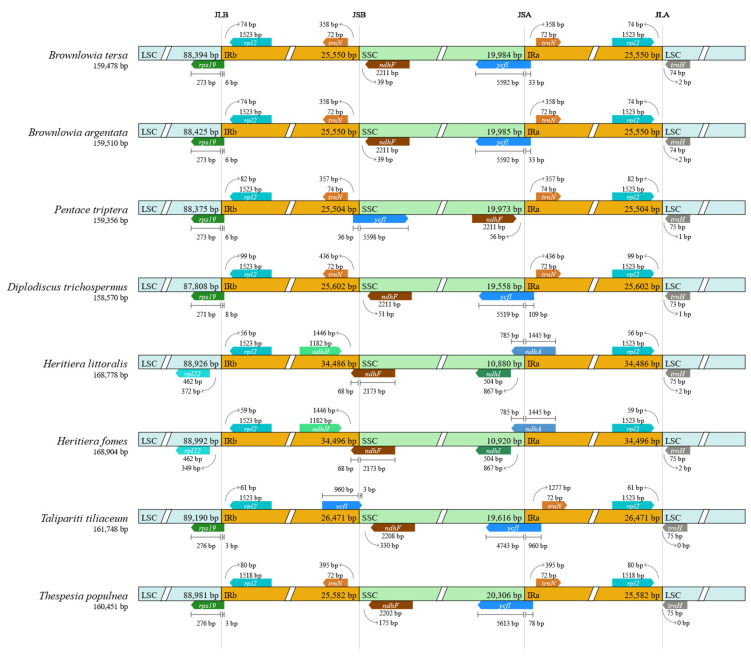
Comparison of the junctions of LSC, SSC, IRa, and IRb regions among eight chloroplast genomes in Malvaceae.

**Figure 4 cimb-47-00074-f004:**
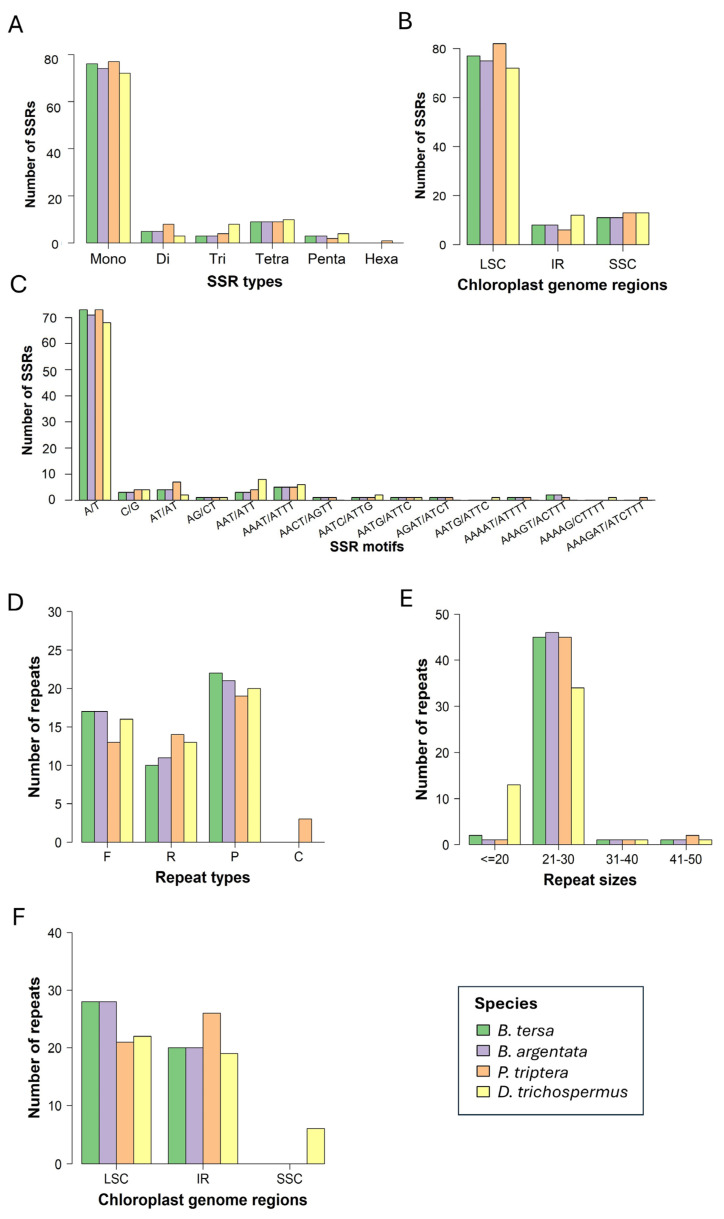
Statistical analysis of simple sequence repeats (SSRs) and dispersed repeats in four chloroplast genomes in Brownlowioideae. (**A**) Sorted by type of SSR. (**B**) Sorted by SSR region of chloroplast genomes. (**C**) Frequency by SSR type. (**D**) Sorted by type of dispersed repeat. (**E**) Frequency by dispersed repeat size. (**F**) Sorted by dispersed repeat region of chloroplast genomes.

**Figure 5 cimb-47-00074-f005:**
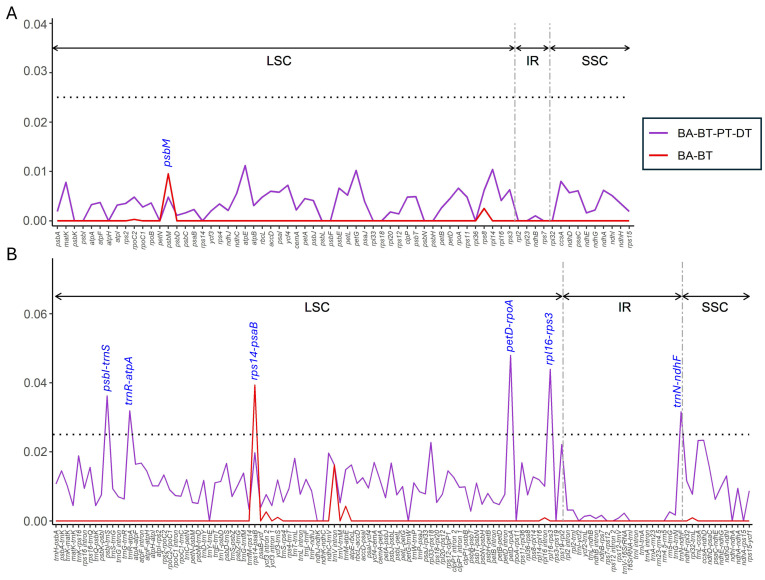
Nucleotide diversity of chloroplast genome sequences of two *Brownlowia* species and two related species in Brownlowioideae. The regions are oriented according to their locations in the chloroplast genomes. (**A**) Coding regions. (**B**) Non-coding regions. BA, BT, PT, and DT represent *Brownlowia argentata*, *Brownlowia tersa*, *Pentace triptera*, and *Diplodiscus trichospermus*, respectively.

**Figure 6 cimb-47-00074-f006:**
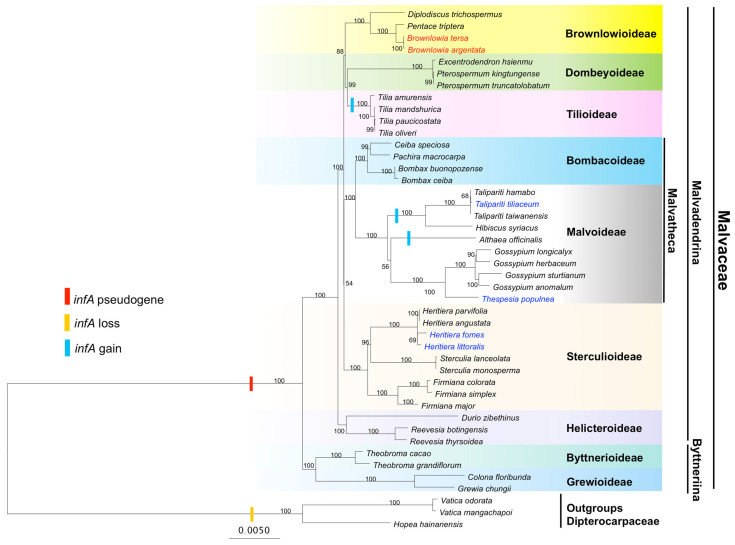
Maximum likelihood (ML) tree for 68 chloroplast protein-coding genes in 44 plant species. Bootstrap values with 1000 replicates show above the branches. The *Brownlowia* species in this study are indicated in red text, while other mangrove species in Malvaceae are indicated in blue text. The Brownlowioideae lineages are indicated in label yellow. Gain, loss, and pseudogene of the *infA* gene are indicated by blue, orange, and red rectangles, respectively.

**Figure 7 cimb-47-00074-f007:**
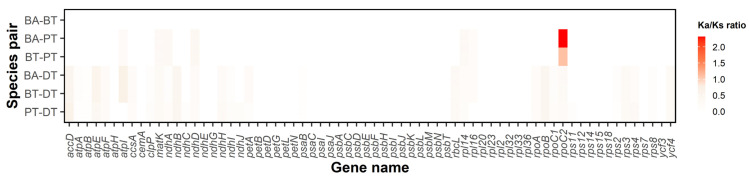
Heatmap of Ka/Ks ratios among six pairwise species in 68 chloroplast genes. The Ka/Ks ratios are shown in the key beside this figure. BA, BT, PT, and DT are *Brownlowia argentata*, *Brownlowia tersa*, *Pentace triptera*, and *Diplodiscus trichospermus*, respectively.

**Table 1 cimb-47-00074-t001:** Chloroplast genome features of two *Brownlowia* species in the subfamily Brownlowioideae.

Characteristics	*Brownlowia tersa*	*Brownlowia argentata*
Genome size (bp)	159,478	159,510
LSC length (bp)	88,394	88,435
SSC length (bp)	19,984	19,985
IR length (bp)	25,550	25,545
GC content (%) Genome	37.05	37.05
LSC	34.89	34.89
SSC	31.35	31.36
IR	43.01	43.01
Coding regions		
No. of total genes	130	130
No. of protein coding genes	84	84
No. of rRNAs	8	8
No. of tRNAs	37	37
No. of duplicated genes in IR	16	16
No. of pseudogenes	1 (*ψinfA*)	1 (*ψinfA*)

**Table 2 cimb-47-00074-t002:** Genes in the chloroplast genome of *B. tersa* and *B. argentata*.

Category	Gene Groups	Gene Name
Photosynthesis	Subunits of photosystem I	*psaA*, *psaB*, *psaC*, *psaI*, *psaJ*
	Subunits of photosystem II	*psbA*, *psbB*, *psbC*, *psbD*, *psbE*, *psbF*, *psbH*, *psbI*, *psbJ*,
		*psbK*, *psbL*, *psbM*, *psbN*, *psbT*, *psbZ*
	Subunits of NADH dehydrogenase	*ndhA* *, *ndhB* * (×2), *ndhC*, *ndhD*, *ndhE*, *ndhF*, *ndhG*,
		*ndhH*, *ndhI*, *ndhJ*, *ndhK*
	Subunits of cytochrome b/f complex	*petA*, *petB* *, *petD* *, *petG*, *petL*, *petN*
	Subunits of ATP synthase	*atpA*, *atpB*, *atpE*, *atpF* *, *atpH*, *atpI*
	Large subunit of rubisco	*rbcL*
Self-replication	Large subunit of ribosome	*rpl2* * (×2), *rpl14*, *rpl16* *, *rpl20*, *rpl22*, *rpl23* (×2), *rpl32*,
		*rpl33*, *rpl36*
	Small subunit of ribosome	*rps2*, *rps3*, *rps4*, *rps7* (×2), *rps8*, *rps11*, *rps12* ** (×2),
		*rps14*, *rps15*, *rps16* *, *rps18*, *rps19*
	DNA-dependent RNA polymerase	*rpoA*, *rpoB*, *rpoC1* *, *rpoC2*
	Ribosomal RNAs	*rrn4.5* (×2), *rrn5* (×2), *rrn16* (×2), *rrn23* (×2)
	Transfer RNAs	*trnA-UGC* * (×2), *trnC-GCA*, *trnD-GUC*, *trnE-UUC*,
		*trnF-GGA*, *trnG-GCC* *, *trnH-GUG*, *trnI-GAC* * (×2),
		*trnK-UUU*, *trnL-CAA* (×2), *trnL-UAA* *, *trnM-CAU* (×2),
		*trnN-GUU* (×2), *trnP-UGG*, *trnQ-UUG*, *trnR-ACG* (×2),
		*trnR-ACG*, *trnR-UCU*, *trnS-GCU*, *trnS-GGA*, *trnS-UGA*,
		*trnT-GGU*, *trnT-UGU*, *trnV-GAC* (×2), *trnV-UAC*,
		*trnW-CCA*, *trnP-UGG*, *trnQ-UUG*
Other genes	Maturase	*matK*
	Protease	*clpP* **
	Envelope membrane protein	*cemA*
	Acetyl-CoA carboxylase	*accD*
	C-type cytochrome synthesis gene	*ccsA*
	Translation initiation factor	*ψinfA*
Genes of unknown	Proteins of unknown function	*ycf1*, *ycf2* (×2), *ycf3* **, *ycf4*

Notes: * Gene with one intron. ** Gene with two introns.

**Table 3 cimb-47-00074-t003:** RNA editing sites of *B. argentata*.

Gene Name	Editing Position in cp Genome	Editing Position in cp Gene	Editing Position in Codon	Codon Change	Amino Acid Change	Editing Efficiency
*matK*	2560	514	1	CAC to UAC	H to Y	47%
	3093	237	2	UCU to UUU	S to F	45%
	3163	214	1	CAU to UAU	H to Y	32%
	3340	155	1	CAC to UAC	H to Y	57%
*atpA*	10,577	472	3	UUC to UUU	F to F	42%
	10,845	383	2	UCA to UUA	S to L	80%
	11,079	305	2	UCA to UUA	S to L	93%
*atpE*	13,321	31	2	CCU to CUU	P to L	93%
*atpI*	15,616	210	2	UCA to UUA	S to L	98%
*rps2*	16,924	83	2	UCG to UUG	S to L	48%
	17,038	45	2	ACA to AUA	T to I	50%
*rpoC1*	23,320	163	2	UCA to UUA	S to L	46%
	24,534	14	2	UCA to UUA	S to L	34%
*rpoB*	27,248	189	2	UCG to UUG	S to L	40%
	27,263	184	2	UCA to UUA	S to L	37%
*psbZ*	37,905	17	2	UCA to UUA	S to L	95%
*rps14*	39,067	50	2	UCA to UUA	S to L	92%
	39,136	27	2	UCA to UUA	S to L	84%
*ndhC*	53,716	108	2	UCA to UUA	S to L	74%
*accD*	62,029	266	2	UCG to UUG	S to L	69%
	62,620	463	2	CCA to CUA	P to L	94%
	62,638	469	2	CCU to CUU	P to L	95%
*psaI*	63,525	28	2	UCU to UUU	S to F	79%
*psbJ*	68,099	20	2	CCU to CUU	P to L	94%
*psbF*	68,484	26	2	UCU to UUU	S to F	86%
*petL*	69,593	2	2	CCU to CUU	P to L	83%
*rps18*	72,056	74	2	UCG to UUG	S to L	16%
*rpl20*	72,481	103	2	UCA to UUA	S to L	19%
*clpP*	73,890	187	1	CAU to UAU	H to Y	95%
*psbN*	78,377	10	2	UCU to UUU	S to F	97%
*petB*	79,693	4	3	GUC to GUU	V to V	79%
	80,099	140	1	CGG to UGG	R to W	95%
*rpoA*	82,180	277	2	UCA to UUA	S to L	45%
	82,810	67	2	UCU to UUU	S to F	23%
*rpl23*	90,239 and 157,707	30	2	UCA to UUA	S to L	31%
	90,257 and 157,689	24	2	UCU to UUU	S to F	58%
*ndhB*	99,260 and 148,686	494	2	CCA to CUA	P to L	78%
	99,486 and 148,460	419	1	CAU to UAU	H to Y	68%
	99,905 and 148,041	279	2	UCA to UUA	S to L	43%
	99,911 and 148,035	277	2	UCG to UUG	S to L	73%
	100,678 and 147,268	249	2	UCU to UUU	S to F	39%
	100,687 and 147,259	246	2	CCA to CUA	P to L	35%
	100,813 and 147,133	204	2	UCA to UUA	S to L	51%
	100,838 and 147,108	196	1	CAU to UAU	H to Y	40%
	100,882 and 147,064	181	2	ACG to AUG	T to M	60%
	100,957and 146,989	156	2	CCA to CUA	P to L	67%
	101,275 and 146,671	50	2	UCA to UUA	S to L	79%
*ndhF*	115,941	97	2	UCA to UUA	S to L	55%
*ndhD*	120,208	437	2	UCA to UUA	S to L	76%
	120,220	433	2	UCA to UUA	S to L	93%
	120,640	293	2	UCA to UUA	S to L	82%
	121,135	128	2	UCA to UUA	S to L	63%
	121,516	1	2	ACG to AUG	T to M	16%
*ndhA*	124,746	189	3	UCA to UUA	S to L	36%
	126,107	114	2	UCA to UUA	S to L	79%

## Data Availability

The data presented in this study are openly available in the NCBI GenBank (https://www.ncbi.nlm.nih.gov/) (accession numbers: PP419968-PP419969).

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
