# Peer review of "Comparative Chloroplast Genomes and Phylogenetic Relationships of True Mangrove Species *Brownlowia tersa* and *Brownlowia argentata* (Malvaceae)"

_cimb, 2025, doi:10.3390/cimb47020074_

Round 1
Reviewer 1 Report
Comments and Suggestions for Authors
The article refers to 'Comparative Chloroplast Genomes and Phylogenetic Relationships of Brownlowia tersa and Brownlowia argentata (Malvaceae), Threatened Mangrove Species'. The article is interesting, but it is need to add some improvements. Please, consider these suggestions:
1) At the end of Introduction, please form general purpose of research. Now, tasks are formed to achieve the purpose, bu what is the purpose?
2) What is research gap? Why this research is need.
3) Description of method is enough, but it will be usuf to add scheme of research to point in simply way what was done.
4) The font in Figure 1 is too small.
5) Also, Figure 5 is not cleaer, please add bigger.
6) There are some limitations of these research? Please add in conclusions.
Author Response
Point 1: At the end of Introduction, please form general purpose of research. Now, tasks are formed to achieve the purpose, but what is the purpose?
Response 1: Thank you for pointing this out. We agree with this comment. So, we have added the purpose of this research in the introduction (Lines 73-76 in the unmasked manuscript).
“To generate genetic resources for Brownlowia species and to reveal evolutionary relationships of Brownlowia species and related species ...”
Point 2: What is the research gap? Why is this research needed?
Response 2: We have added the research gap in the introduction (Lines 72-73).
“Recently, no study has focused on the genome sequence of Brownlowia species, which is important for studying evolution and species identification.”
Point 3: Description of method is enough, but it will be used to add a scheme of research to point in simply way what was done.
Response 3: We have created a pipeline for assembling and analyzing chloroplast genomes of Brownlowia tersa and Brownlowia argentata in figure S1. We have also added it in the revised manuscript (Line 101).
Point 4: The font in Figure 1 is too small.
Response 4: We have modified figure 1 to improve its quality.
Point 5: Also, Figure 5 is not clear, please add bigger.
Response 5: We have edited figure 5.
Point 6: Are there some limitations of this research? Please add conclusions.
Response 6: We have added one limitation of this research in the conclusion section (Lines 529-530).
“Notably, the chloroplast genomes may not be efficient markers for DNA barcoding in Brownlowia mangrove species due to high genetic similarity.”

Reviewer 2 Report
Comments and Suggestions for Authors
The document is adequate, original and well supported, however the following recommendations are recommended
Line 72 eliminate "in this study" as well as throughout the text so that the writing is more impersonal.
Line 86 increase the description of the method of handling plant material
Line 211 Improve the quality of figure 1 since it makes reading and interpretation difficult
Line 412, in this section it is not discussed whether there is any relevance of the pseudogenes previously found, if not, make it clear.
Line 522 the conclusions are a summary of the results. It is necessary to write the importance of the study with something more precise than what was written in the last sentence.
Author Response
Point 1: Line 72 eliminates "in this study" as well as throughout the text so that the writing is more impersonal.
Response 1: Agree. We have revised the sentence (Lines 73-76 in the unmasked manuscript)
Point 2: Line 86 increases the description of the method of handling plant material.
Response 2: We have revised the part of plant material to describe the method of handling plant material (Lines 87-92)
Point 3: Line 211 Improve the quality of figure 1 since it makes reading and interpretation difficult
Response 3: We have edited figure 1 to improve its quality.
Point 4: Line 412, in this section it is not discussed whether there is any relevance of the pseudogenes previously found, if not, make it clear.
Response 4: We have revised this section (Lines 414-415)
Point 5: Line 522 the conclusions are a summary of the results. It is necessary to write the importance of studying with something more precise than what was written in the last sentence.
Response 5: We have revised the conclusion, especially the last sentence (Lines 536-538)

Reviewer 3 Report
Comments and Suggestions for Authors
Dear Authors, I have a few comments.
The title, abstract and Introduction are sufficiently described. However, the title could be a little changed: you use in title ,,threatened” but in the Introduction line ,,near threatened” and the risk will be in the future (Lines 50-54).
The Methods are clearly and sufficiently.
The Figure 1, 5, 7 are unreadable, the Table 1 is incomprehensible.
The Latin names of the species should be in italics.
The Results, Discussion, Conclusions – are sufficiently.
Author Response
Point 1: The title could be a little changed: you use in title, threatened” but in the Introduction line ,near threatened” and the risk will be in the future (Lines 50-54).
Response 1: We have revised the title as “Comparative Chloroplast Genomes and Phylogenetic Relationships of Brownlowia tersa and Brownlowia argentata (Malvaceae), True Mangrove Species”.
Point 2: The Figure 1, 5, 7 are unreadable, the Table 1 is incomprehensible.
Response 2: We have edited figures 1,5 and 7 as well as table 1.
Point 3: The Latin names of the species should be in italics.
Response 3: Agree. We have edited all species names in italics.
